# Generating QM1B with PySCF$_{\text{IPU}}$

**Alexander Mathiasen[1]**   **Hatem Helal[1]**   **Kerstin Klaser[1]**   **Paul Balanca[1]**   **Josef Dean[1]**
**Carlo Luschi[1]**   **Dominique Beaini[2, 3, 4]**   **Andrew Fitzgibbon[1]**   **Dominic Masters[1]**

[1]Graphcore    [2]Mila - Québec AI Institute    [3]Valence    [4]Université de Montréal

## Abstract

The emergence of foundation models in Computer Vision and Natural Language Processing have resulted in immense progress on downstream tasks. This progress was enabled by datasets with billions of training examples. Similar benefits are yet to be unlocked for quantum chemistry, where the potential of deep learning is constrained by comparatively small datasets with 100k to 20M training examples. These datasets are limited in size because the labels are computed using the accurate (but computationally demanding) predictions of Density Functional Theory (DFT). Notably, prior DFT datasets were created using CPU supercomputers without leveraging hardware acceleration. In this paper, we take a first step towards utilising hardware accelerators by introducing the data generator PySCF$_{\text{IPU}}$ using Intelligence Processing Units (IPUs). This allowed us to create the dataset QM1B with one billion training examples containing 9-11 heavy atoms. We demonstrate that a simple baseline neural network (SchNet 9M) improves its performance by simply increasing the amount of training data without additional inductive biases. To encourage future researchers to use QM1B responsibly, we highlight several limitations of QM1B and emphasise the low-resolution of our DFT options, which also serves as motivation for even larger, more accurate datasets. Code and dataset.

## 1   Introduction

Artificial Intelligence research is quickly moving towards building systems that are able to perform generally applicable functions using foundation models that are pretrained on billions of training examples. Foundation models have transformed natural language processing (NLP) [1] and computer vision (CV) [2], but have not yet been demonstrated in the important field of molecular machine learning. Addressing this has the potential to accelerate the design of new medicines and materials.

A promising approach within molecular machine learning is to train neural networks (NN) to approximate the predictions of quantum chemistry, yielding 1000x faster predictions [3, 4] with errors approaching that of experimental measurement. However, current molecular datasets are limited to 100k-20M training examples, that are seemingly insufficient to train foundation models. Due to the high cost and time consuming nature of constructing chemical datasets directly from experimental measurement, prior datasets rely on computational methods to generate labels. The computational method routinely used to generate machine learning datasets is Density Functional Theory (DFT). Although DFT strikes a favourable balance between accuracy and computational cost relative to other molecular simulation methods, its computational cost has nevertheless limited prior datasets[1] to fewer than 20M training examples. Notably, prior DFT datasets such as QM9 [6, 7], ANI-1 [8] or PCQ [9] were created by running DFT on CPUs. To the best of our knowledge no prior work utilises hardware acceleration to create larger DFT datasets for deep learning.

---

[1]Gaussian basis DFT datasets. This excludes semi-empirical methods [5] and plane-wave DFT [4].

37th Conference on Neural Information Processing Systems (NeurIPS 2023) Track on Datasets and Benchmarks.

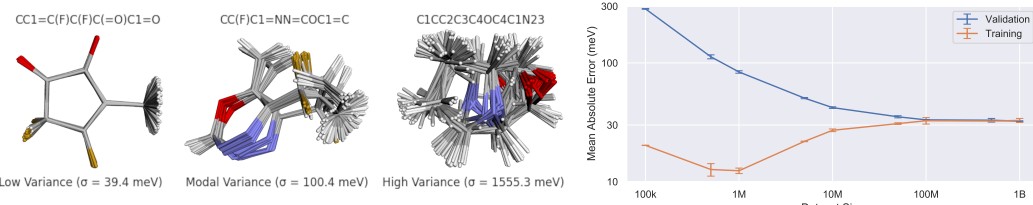

Figure 1: We took 1.09M molecules with $\geq 9$ atoms from GDB11 [11, 12], and used PySCF$_{\text{IPU}}$ to compute molecular properties (e.g. HOMO-LUMO (HL) gap) for a set of up to 1000 conformers. Three such sets are visualised here, corresponding to low, modal, and high variance of HL gap. Training a SchNet 9M model to predict HL gap shows improvement as the number of training samples approaches 500M. All training runs see $10^9$ examples, sampling with replacement. We conjecture that billions of samples will enable the training of molecular foundation models.

We introduce PySCF$_{\text{IPU}}$, a DFT data generator which utilises Intelligence Processing Units (IPUs [10]) to accelerate molecular property dataset generation. We used PySCF$_{\text{IPU}}$ to generate QM1B with one billion DFT training examples within 40000 IPU hours, substantially less than the two years it took to create PCQ on CPU supercomputers. The larger scale was enabled by several changes in the dataset generation which we outline in Section 4. The most important change, from a machine learning perspective, is that we increased the number of training examples by reducing the DFT accuracy. We advise that QM1B is only used for research purposes and discourage applications in deployed systems without carefully investigating the consequences of the reduced DFT accuracy. It remains to be found how pretraining on 1B low-resolution DFT training examples may bias a NN which is subsequently fine-tuned on downstream tasks. We encourage future research to investigate how DFT options bias subsequent neural networks by jointly iterating on dataset creation and model training. To enable such research we do not only release QM1B but also the entire software stack used to generate QM1B.

To investigate whether more training data improves neural networks, we trained a baseline SchNet with 9M parameters on differently sized subsets of QM1B. We found the validation Mean Absolute Error (MAE) improved from 285meV to 32meV, see Figure 1. Notably, the training and validation MAE overlap at the larger scales, indicative of underfitting, paving the way for future work to train large foundation molecular neural networks.

After pre-training SchNet on QM1B we fine-tuned SchNet on QM9. Compared to training SchNet on QM9 from scratch, we found this to improve validation MAE from 54.13meV to 30.2meV.

The contributions of this work can be summarised as follows:

1. We present PySCF$_{\text{IPU}}$, a new hardware accelerated DFT data generator for deep learning.
2. We release the dataset QM1B, a low-resolution DFT dataset with $10^9$ training examples.
3. We train a simple NN baseline which improves performance when trained on more data, validating the scale of QM1B.

## 2  Related Work

The core contributions of this work touch on three mature fields that have typically been studied in isolation. We provide a brief history of DFT software packages and follow this with their application to generating large quantum chemical datasets. Finally, we summarise efforts to approximate DFT with machine learning models.

### 2.1  DFT Libraries

A suitable approximation to quantum mechanics can provide a universal framework for predicting properties of molecules and materials [13]. The most widely applied approximation to quantum theory is Kohn-Sham DFT [14] which is implemented in a number of widely available DFT packages. These packages have evolved over decades and have successfully adapted to the constantly shifting landscape of high-performance computing. Notably, a substantial fraction of nation-level supercomputers are

dedicated to running DFT simulations, e.g., 30% of the US National Energy Research Scientific Computing Center (NERSC) and up to 40% of the UK ARCHER2 supercomputer [15, 16]. We highlight three advances in scientific computing that have accelerated progress in machine learning research that have yet to be widely adopted by DFT software libraries:

1. Ubiquitous hardware acceleration using low-precision numerical formats.
2. Tensor computations in a high-level API like NumPy [17].
3. Automatic differentiation [18].

Crucially, the core algorithm development of many widely used DFT packages predates the rise of NumPy and the Python ecosystem. This also coincides with the increased availability of specialised hardware accelerators for scientific computing. For this reason, most DFT libraries are still developed in languages like FORTRAN and C, without support for automatic differentiation. Hardware acceleration in this low-level programming environment is limited, with marginal performance gains reported [19]. One notable exception is the commerical TeraChem library [20] which was engineered specifically for GPUs.

Table 1: Comparison of different Gaussian basis set DFT libraries frequently deployed on high performance computing clusters[2]. Despite the fact that there are many DFT libraries that support hardware acceleration, we know of none that have been used to generate machine learning datasets.

| Library | Autodiff | Hardware | Precision | License | Language |
|---------|----------|----------|-----------|---------|----------|
| Gaussian9 [21] | no | CPU/GPU | float64 | Commercial | - |
| TeraChem [20] | no | CPU/GPU | mixed float32/64 | Commercial | C++ |
| Psi4 [22] | no | CPU/GPU | float64 | Open Source | C++ |
| GAMESS [23] | no | CPU/GPU | float64 | Open Source | C, Fortran77 |
| PySCF [24] | no | CPU | float64 | Open Source | NumPy, C |
| PySCF$_{IPU}$ (ours) | yes* | IPU | float32 | Open Source | Jax, C |

## 2.2 DFT Datasets

The authors of previous DFT datasets made different choices, e.g., basis set and number of conformers. We list several examples in Table 2 limited to Gaussian basis DFT. [3] It is not clear how these choices impact subsequent deep learning models. We hope PySCF$_{IPU}$ will enable future researchers to jointly iterate on dataset creation and model training. While prior datasets use DFT libraries with some support for hardware acceleration, we found all dataset papers relied on CPUs.

Table 2: Comparison of different DFT datasets.

| Dataset | Graphs | Conformers | Basis Set | XC | Atoms | Time | Library | Hardware |
|---------|--------|------------|-----------|-----|-------|------|---------|----------|
| QM9 [7] | 133.9k | 133.9k | 6-31G(2df,p) | B3LYP | $\leq 9$ | - | ? | CPU |
| PCQ [9] | 3.38M | 3.38M | 6-31G* | B3LYP | $\leq 20$ | ~2 years [25] | GAMESS | CPU |
| SPICE [26] | 19238 | 1.13M | def2-TZVPPD | $\omega$B97M-D3(BJ) | 2-96 | - | PSI4 | CPU |
| ANI-1 [27] | 57462 | 20M | 6-31G(d) | $\omega$B97x | $\leq 8$ | - | Gaussian9 | CPU |
| QM1B | 1.09M | 1B | STO-3G | B3LYP | 9-11 | $\sim$ 5 days[4] | PySCF$_{IPU}$ | IPU |

## 2.3 Neural Net Approximations to DFT

Similar to CV and NLP, molecular machine learning has seen substantial effort towards optimising neural networks to better approximate DFT. Broadly speaking, these approaches include denoising and auxilliary tasks [28, 29, 30, 31, 32], pretraining/finetuning [30, 33, 34, 35, 36], positonal encodings [37, 38, 39, 40, 41] and architectural improvments [42, 43, 44]. Some work concentrated on performing equilibrium based predictions with 2D molecular representations bypassing expensive DFT-based structure optimisations [45, 46, 47, 48, 49]. [4] investigated how NNs improve their DFT

---

[2]This excludes e.g. the unpublished github repository gpu4pyscf.

[3]This excludes plane-wave DFT as used in [4] and semi-empirical methods as used in [25].

[4]Estimated as 40000 IPU hours using 320 IPUs.

approximation while increasing training examples. They concluded that current methods would require orders of magnitude more training examples to solve their catalysis task. For demonstration purposes, we chose to scale one of the simplest models, SchNet [50], to investigate how far performance can be improved by simply scaling training data without incorporating additional architectural innovations. In Table 3 we compare SchNet trained on QM9 against a selection of other methods that were trained from scratch (i.e. no pretraining/finetuning). We point out that the parameter count is orders of magnitude lower than foundation models in CV and NLP. Furthermore, the machine learning errors around 20-70meV are orders of magnitude larger than the convergence thresholds commonly employed in DFT. This motivated us to challenge the common assumption that DFT requires double precision (float64) for the special case of generating training data for machine learning.

Table 3: Results of previous Neural Networks trained from scratch on the DFT dataset QM9.

| Method | # Params | MAE QM9 HL gap (meV) | MAE QM9 energy (meV) |
|---|---|---|---|
| SchNet [50] | 432k | 63 | 14.6 |
| PaiNN [51] | 589k | 45.7 | **5.85** |
| DimeNet++ [52] | 1.8M | 32.6 | 6.32 |
| GNS + Noisy Nodes [31] | - | **28.6** | 7.3 |

## 3 PySCF$_{\text{IPU}}$

The Python-based Simulations of Chemistry Framework (PySCF) implements multiple Self-Consistent Field methods (SCF) including DFT. Importantly, relative to many other DFT libraries, PySCF is free to use, open-source[5] and implemented mainly in Python and NumPy with a few of the computational demanding operations written in C. Furthermore, PySCF supports a plethora of DFT options, enabling it to reproduce quantum chemical properties as computed in machine learning datasets like QM9 or PCQ. All of the above made PySCF a good starting point to port for IPUs.

Intelligence Processing Units (IPUs) are a type of accelerated hardware designed specifically for machine learning workloads. We found IPUs allowed us to speed up DFT data generation due to two reasons:

1. IPUs have 940MB on-chip memory with 12-65TB/s bandwidth, enough to perform small DFT computations without relying on off-chip RAM with < 3TB/s bandwidth.
2. IPUs support Multiple Instruction Multiple Data (MIMD) parallelism which simplifies parallel computation of the notoriously difficult computation of Electron Repulsion Integrals.

### 3.1 DFT Primer.

Our data generator PySCF$_{\text{IPU}}$ implements DFT. To allow machine learning researchers to reason about our data generator, this section describes *what* Gaussian basis set DFT implementations like PySCF$_{\text{IPU}}$ do. We deliberately limit ourselves to explaining *what* DFT do and exclude explaining the underlying physics, e.g., under which assumptions DFT can be derived from the Schrödinger equation (see instead [53] or [54, Appendix B]).

The inputs to a DFT computation are atomic positions and atomic numbers. These inputs are used to compute a matrix $\mathbf{V}$ which represents the system of interest. One then solves the following matrix equation for $(\mathbf{C}, \boldsymbol{\epsilon})$.

$$\big[\mathbf{V} + \mathbf{T} + J(\rho(\mathbf{C})) + K(\rho(\mathbf{C})) + V_{XC}(\rho(\mathbf{C}))\big]\mathbf{C} = \mathbf{S}\mathbf{C}\boldsymbol{\epsilon}$$

$$\rho(\mathbf{C}) = \mathbf{C}\mathbf{D}\mathbf{C}^T = \boldsymbol{\rho} \quad D_{ij} = \begin{cases} 2 & \text{if } i = j \le N_{\text{electrons}/2} \\ 0 & \text{else} \end{cases}$$

$$J(\boldsymbol{\rho})_{kl} = \sum_{ij} I_{ijkl}^{2e}\rho_{ij} \quad K(\boldsymbol{\rho})_{il} = -\frac{1}{2}\sum_{jk} I_{ijkl}^{2e}\rho_{jk}$$

Here $\mathbf{I}^{2e} \in \mathbb{R}^{N \times N \times N \times N}$ while $\{\boldsymbol{\rho}, \mathbf{V}, \mathbf{T}, \mathbf{C}, \mathbf{S}, \boldsymbol{\epsilon}\}$ are in $\mathbb{R}^{N \times N}$, $(\mathbf{D}, \boldsymbol{\epsilon})$ are diagonal matrices and $\{\rho, J, K, V_{XC}\}$ are functions with $(N, N)$ matrices as input and output. Given a physical system,

---

[5]Under the Apache-2.0 License under which we plan to open-source PySCF$_{\text{IPU}}$.

the DFT algorithm proceeds in iterations. At iteration 0, we precompute $(\mathbf{V}, \mathbf{T}, \mathbf{S}, \mathbf{I}^{2e})$ and initialise $\mathbf{C}_1$. At iteration $i$, we compute $(\boldsymbol{\rho}_i = \rho(\mathbf{C}_i), V_{XC}(\boldsymbol{\rho}_i), J(\boldsymbol{\rho}_i), K(\boldsymbol{\rho}_i))$, and then solve the above equation for $\mathbf{C}_{i+1}$ (which corresponds to a generalised eigenproblem). If the iterations converge to self-consistency, one attains a solution $(\mathbf{C}, \boldsymbol{\epsilon})$ from which several molecular properties can be computed, e.g.

$$\text{HL gap} := \epsilon_{HOMO} - \epsilon_{LUMO}$$

DFT has two well-known trade-offs between accuracy and compute time:

1. $V_{XC}$ attempts to correct the approximation error between DFT and the Schrödinger equation; hundreds of corrections exist such as LDA [13], B3LYP [55, 56] and $\omega$B97x [57].
2. $\mathbf{C}$ represents molecular orbitals as a linear combination of Gaussian functions; many such Gaussian basis sets have been extensively studied [58].

Prior work trained neural networks to improve the inherent errors from DFT for $V_{XC}$ [59, 60] and the representation of molecular orbitals [54].

### 3.2 PySCF IPU Implementation

To utilise IPUs we ported PySCF from NumPy to JAX [61], which can target IPUs by using IPU TensorFlow XLA backend [62]. This left two remaining parts in C: [6]

1. LIBXC [64] which computes $V_{XC}$
2. LIBCINT [65] which computes $(\mathbf{T}, \mathbf{V}, \mathbf{S}, \mathbf{I}^{2e})$

**Libxc.** We only support the B3LYP functional. The energy computation of B3LYP was implemented in JAX, which allowed us to use JAX autograd to compute the B3LYP potential. We use float32 instead of float64. JAX-XC [66] recently machine translated the libxc Maple files to JAX. This may allow us support more functionals by extending JAX-XC to float32.

**Libcint.** We implemented the computation of $\mathbf{I}^{2e}$ by porting the libcint implementation of the Rys Quadrature algorithm from C to the IPU. The code implements a function INT2E which is called many times with different inputs. Each call can be done in parallel. However, the different calls perform different computations making it tricky to parallelise using Single Instruction Multiple Data (SIMD). In contrast, using the IPUs MIMD parallelism all 8832 IPU threads can independently compute one call to INT2E in parallel. $\mathbf{I}^{2e}$ satisfies an 8x symmetry[7] which, when exploited, (1) reduces bytes needed to store $\mathbf{I}^{2e}$ by 8x and (2) gives an 8x reduction in FLOPs needed to compute $K(\boldsymbol{\rho}, \mathbf{I}^{2e})$ and $J(\boldsymbol{\rho}, \mathbf{I}^{2e})$ (usually computed using np.einsum). To utilise the 8x symmetry we implemented a custom einsum algorithm. The memory usage of our current implementation can be improved as outlined in Section 5.

**Numerical Error of PySCF$_{\text{IPU}}$.** DFT libraries usually use double precision (float64), see Table 1. In contrast, NNs are usually trained in float32 to decrease compute time, e.g., some hardware accelerators can perform matrix multiplications more than 10x faster in float32 than float64. For the particular case of libcint, we found float32 to be around 6x faster than float64, likely due to IPUs ability to hardware accelerate float32 instead of simulating float64 in software (similar to the 10x reported by [67]). While float32 has yet to be thoroughly demonstrated in DFT libraries, our goal is to generate data to train NNs, therefore, our requirements for numerical precision is less stringent compared to mainstream DFT libraries, our errors just have to be below that of NNs (20-70meV for HL gap). To investigate our numerical errors, we recomputed DFT with PySCF in float64 for 10k SMILES strings from QM1B. This allowed us to compute a the mean absolute error (MAE) of PySCF$_{\text{IPU}}$ relative to PySCF. For converged[8] molecules our MAE was 6meV for energies and 0.2meV for HL gap, see Figure 2. Our numerical error for HL gap is thus 100x smaller than the errors achieved by neural networks (but similar for energy), see Table 3.

---

[6][63] cleverly extends PySCF with AD by using JAX to pair all C calls with gradient C calls (only for CPUs).

[7]$\mathbf{I}^{2e}_{ijkl} = \mathbf{I}^{2e}_{ijlk} = \mathbf{I}^{2e}_{jikl} = \mathbf{I}^{2e}_{jilk} = \mathbf{I}^{2e}_{jilk} = \mathbf{I}^{2e}_{klij} = \mathbf{I}^{2e}_{klji} = \mathbf{I}^{2e}_{lkij} = \mathbf{I}^{2e}_{lkji}$.

[8]Converged is defined as np.std(energies[-5:])<0.01. Around 99% converged. In the future the remaining 1% could be added using PySCF for a negligble cost.

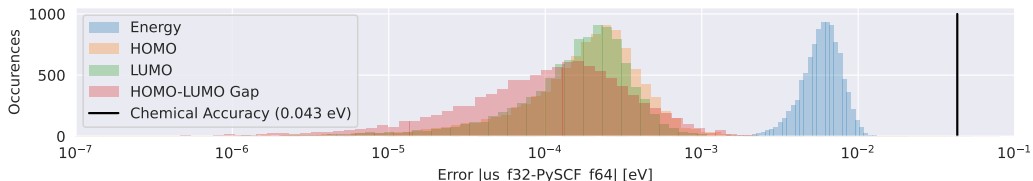

Figure 2: Numerical error of PySCF$_{\text{IPU}}$ in float32 compared to PySCF in float64. We suspect the dfference between HL gap and energy is caused by energies being 100x larger than HL gaps.

**Timing PySCF$_{\text{IPU}}$.**   Our logs recorded 38729 IPU hours generating QM1B. We estimate less than $4\%$ was spent compiling. We completed the data generation within 5 days, by utilizing a varying number of 256-384 IPUs split over 16-24 POD16s. Each IPU POD16 has two physical EPYC CPUs with 240 vCPUs. On a POD16, for the example molecule FC=C1C2CN2N=C1C=O, we can do 82 DFTs/sec with PySCF and 228 DFTs/sec with PySCF$_{\text{IPU}}$, see Figure 3 for a profile of PySCF$_{\text{IPU}}$.

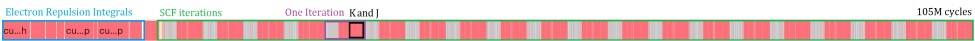

Figure 3: Profile of our DFT computation for "Fc1nocnc(=O)c1=O".

### 3.3   Limitations

PySCF$_{\text{IPU}}$ is an ongoing research project and thus has several limitations:

1. The 940MB limits PySCF$_{\text{IPU}}$ to $\leq 12$ heavy atoms with a less accurate electron representation (STO-3G).
2. Larger numerical errors due to float32 instead of float64.
3. The first simulation of a molecule with $N$ atomic orbitals takes approximately five additional minutes due to the ahead-of-time compilation model used for IPUs (for QM1B we estimate spending less than $4\%$ time compiling).

Our current implementation supports restricted Kohn-Sham, B3LYP functional [55, 56], no inter-atomic force and the basis sets $\{$STO3G, 6-31G$\}$.

## 4   QM1B

We created QM1B by taking 1.09M SMILES strings from the Generated Data Bank (GDB)[9] with 9-11 heavy atoms (GDB11). Hydrogen atoms were added by using RDKIT resulting in 0 to 11 Hydrogen atoms. We subsequently generated up to 1000 conformers for each molecule using the ETKDG algorithm [68] as implemented in RDKit [69]. This led to a total of 305.8M, 568,7M and 205.4M conformers for 9,10,11 heavy atoms respectively. We then computed the following properties on the resulting 1B conformers using PySCF$_{\text{IPU}}$:

1. Energy
2. Highest Occupied Molecular Orbital (HOMO) Energy
3. Lowest Occupied Molecular Orbital (LUMO) Energy

**Conformers.**   Datasets like QM9 and PCQ use DFT to compute atom positions for which each molecule is in equilibrium, i.e., the forces acting on all atoms are zero:

$$\mathbf{F} = -\frac{\partial \ \text{DFT\_energy(atom\_positions)}}{\partial \ \text{atom\_positions}} = \mathbf{0} \in \mathbb{R}^{\text{num\_atoms} \times 3}.$$

Equilibrium atom positions are computed through structure optimization, typically done by minimizing energy with respect to atom positions using standard optimisation methods such as conjugate gradients or LM-BFGS. This takes $\approx 50$ iterations of LM-BFGS where each iteration use DFT to

---

[9]GDB is released under the Creative Commons Attribution 4.0 International. We received written consent from the author of GDB use it for QM1B.

compute **F**. One can thus get 50x more examples of DFT by evaluating DFT on atom positions that are not structure optimised (similar in spirit to ANI-1 [8]). By utilising the ETKDG algorithm to generate conformers we get 50x more data, while relying on the ETKDG algorithm to explore the conformational space. We investigate the diversity of conformers generated using this method in Figure 4(a), which depicts the distribution of the standard deviation of each molecule's HL Gap over all conformers. Figure 4(b) shows the distribution of conformers of a few molecules from the extremes of the dataset, and visualises several different conformers for each molecule super-imposed on top of each other.

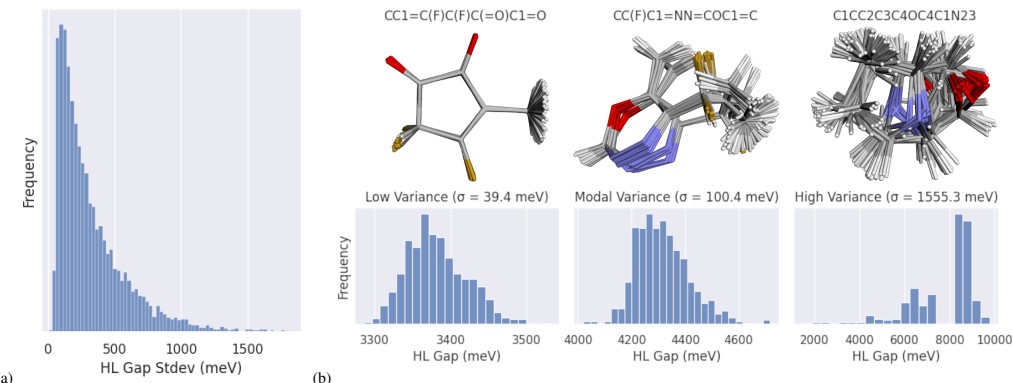

Figure 4: (a) If we compute the standard deviation of HL gaps over all conformers for a single SMILES string, then this figure plots the distribution of those standard deviations over all SMILES strings in the dataset. (b) The distribution of HL gaps across different conformers for 3 example molecules. These examples are chosen from the left tail, the peak, and the right tail of the distribution of all HL gap variances seen in (a).

**Lower Resolution.** For a fixed compute budget to generate data for machine learning, we can think of the DFT options as trading-off dataset size for data quality (DFT accuracy). Relative to prior datasets, we chose a less accurate DFT to generate more training examples. To compare the resulting inaccuracies, we compare the HL gap computed for the same molecule with different options. In particular, Figure 5 compares our STO-3G/B3LYP options against 6-31G*/B3LYP options (attempting to mimic the options from PCQ as closely as possible). The mean absolute error between the options are 360meV, much larger than chemical accuracy (43meV) or the errors attained by SchNet (63meV). It remains an open problem how pretraining on 1B such low-resolution conformers may bias a Neural Network subsequently fine-tuned on a smaller but more accurate dataset like PCQ. As a preliminary investigation, we pre-trained SchNet on QM1B and then finetuned it on QM9. Compared to only training SchNet on QM9, fine-tuning improved performance from 54.13meV to 30.2meV.

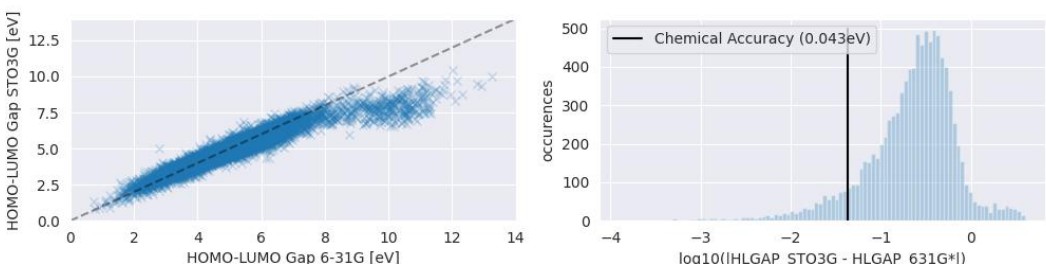

Figure 5: Comparison of different DFT options for 10k molecules from QM1B. Notably, the disagreement between the DFT options are larger than chemical accuracy. It remains to be found how pretraining on 1B molecules with "lower resolution DFT" biases subsequent models fine-tuned on downstream tasks.

**KDEplot of HL Energy Gap.**   To compare HL gap prediction tasks across different datasets we plotted the kernel density estimate (KDE) of the HL gaps. We plot these in Figure 6 for QM9, PCQ and QM1B. To further investigate the consequences of our lower resolution DFT, we also compared our DFT implementation with the optimised atom positions from QM9 against another version of QM9 with atom positions from ETKDG [68] as implemented in RDKit [69].

**Basis Sets and Neural Network.**   While Figure 5 visualises the disagreement between STO3G and 6-31G*, it remains unclear how the disagreement impacts the training behaviour of neural networks. To investigate this we created a version of QM9 with the same DFT options as used for QM1B while using the same atom positions. This allowed us to train the same SchNet model on the same molecules, but with labels computed using different DFT options. Figure 7 shows that the resulting loss curves broadly overlap throughout training.

We also investigated the impact of using off-equilibrium molecules through the RDKit conformers, that is, we recomputed QM9 but using atom positions from RDKit conformers. We found the resulting loss curves were worse (larger MAE), suggesting off-equilibrium prediction may be a fundamentally harder task. We emphasise further experiments are needed to gain certainty, and point out that PySCF$_{IPU}$ may allow future research to investigate these types of molcular dataset biases.

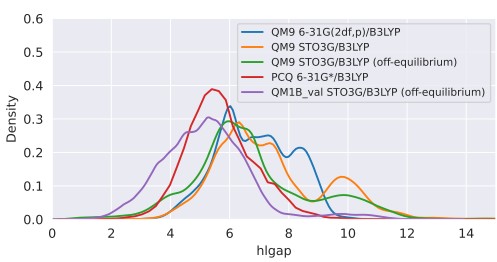

Figure 6: Kernel Density Estimate (KDE) plot of the HL gaps in different datsaets.

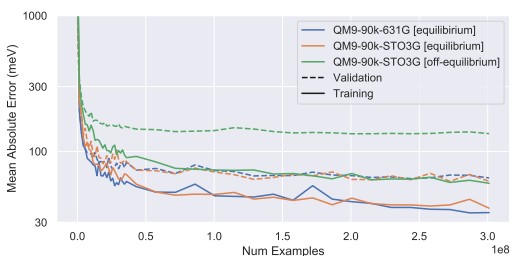

Figure 7: The SchNet 9M model trained on three different versions of QM9 to investigate how choices in dataset creation impact training.

**Scaling SchNet Baseline on QM1B.**   We trained a 9M parameters SchNet model on varying size subsets of the full QM1B dataset. Each subsets consists of N molecule and conformer pairs sampled uniformly from QM1B. For the full dataset a single epoch of training is performed with the number of epochs for each smaller subset adjusted to keep the total number of iterations fixed. The results are presented in Figure 1 and report mean values for three independent seeds. They show that the validation MAE improves as more samples are added to the dataset, as expected. Interestingly we see that the training and validation performance plateaus after approximately 100M samples are in the dataset, this suggests that the model is underfitting, and increasing the model size would further improve performance.

**Application Scenario: Pre-training on QM1B, fine-tuning on QM9.**   After pre-training the 9M SchNet on QM1B we fine-tuned it QM9. Compared to trainined the 9M SchNet on QM9 from scratch, pretraining (and subsequent fine-tuning) improved validation performance from 54.13meV to 30.2meV. To avoid data leakage, we restricted fine-tuning to a subset of 88k molecules from QM9 for which their SMILES strings did not appear in the version of GDB used to create QM1B. For discussion of pre-training tasks see Appendix A.

## 5   Future Work: Larger Molecules and Higher Resolution DFT

The main limitation our of implementation is the fact that it runs out of memory when $N > 70$, where $N$ is the number of atomic orbitals. This limited us to use $\leq 12$ atoms in STO-3G with a small grid size (10-30k).

**Current (Naive) Bottleneck.**   We can view $\mathbf{I}^{2e}$ as a matrix $M \in \mathbb{R}^{N^2, N^2}$ so $K(\boldsymbol{\rho}) = Mx$ where $x = \text{flatten}(\boldsymbol{\rho})$. For $N = 70$ we spend 173MB computing $v = Mx$. We represent $M$ with the 8x

symmetry as a list of matrices of size (num_integrals, integral_size). $M$ is stored distributed over all the cores inside one IPU and thus never needs to move. We compute $v_i = Mx$ by copying $x$ for every thread allowing us to compute $x_i = M_{thread_i} x$ where $M_{thread_i}$ is the part of $M$ stored on core $i$. The result is then $v = \sum_i x_i$. This uses $N^2 \cdot$ number_of_threads floats (or 173MB). The memory consumption can be reduced to 173MB/N=2.47MB by splitting $x$ into $N$ batches each with $N$ floats. The current strategy was a stepping stone, the main advantage is ease of implementation. Notably, prior work disregarded this 8x symmetry at the cost of a 5-8x slowdown [63].

**Recomputation.**    Instead of precomputing $\mathbf{I}^{2e}$ we can recompute the entries of $\mathbf{I}^{2e}$ whenever needed during the simultaneous computation of $K(\boldsymbol{\rho}, \mathbf{I}^{2e})$ and $J(\boldsymbol{\rho}, \mathbf{I}^{2e})$. For the case of Figure 3(a) it took 15.3M out of 105M cycles to compute $\mathbf{I}^{2e}$. Recomputing $\mathbf{I}^{2e}$ all 20 iterations would increase cycle count from 105M to an estimated 395.7M cycles (57.5ms to 216.8ms on a 1.825GHz Mk2 BOW). Finally, if we also recompute the evaluation of the atomic orbitals on the XC grid our memory consumption becomes $N^2 + \text{grid\_size} \cdot 4 + B$, where $B$ is the memory used while (re)computing $K(\boldsymbol{\rho}, \mathbf{I}^{2e})$ and $J(\boldsymbol{\rho}, \mathbf{I}^{2e})$ (independent of $N$).

**Using Multiple IPUs.**    We exemplify our plan to parallelise over 15GB SRAM in 16 IPUs following Figure 3(a). The computation of $\mathbf{I}^{2e}$ (Eletron Repulsion Integral) can be split over 16 IPUs in the same way we already utilise MIMD parallelism to split them over the 8832 threads. The computation of $K$ and $J$ can also be split over 16 IPUs with one REDUCE_SUM of size $N^2$. The remainder of the computation can be repeated independently on each IPU.

## 6  Discussion.

Prior work usually falls in one of the following three categories: (1) DFT libraries, (2) DFT datasets, and (3) neural networks trained on DFT datasets. This article touches on all three, demonstrating how choices made for DFT libraries and dataset generation (1,2) can impact neural network training (3). As molecular machine learning turns towards training large foundation models, we hope our work will help generate sufficiently large datasets with billions (or trillions) of molecules. Furthermore, we hope to allow researchers to generate datasets for downstream tasks used solely to fine-tune such foundation models.

# 7 Broader Impact.

Our work attempts to improve neural network approximations to Density Funtional Theory (DFT) by generating larger datasets. Improving DFT holds the promise to accelerate the development of new medicines and materials, with positive implications for health outcomes and global warming. However, the same technology may someday also allow bad actors to design pathogens, viruses or weapons.

## Acknowledgments and Disclosure of Funding

We thank Graphcore research for help and feedback through the entire project. In particular, a huge thanks to Charlie Blake for help with numerical analysis and beautiful visualizations. We also thank Ivan Franzoni, Hadrien Mary Prudencio Tossou, Therence Bois and Cristian Gabellini for valuable feedback. The sole funder of this project was Graphcore.

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

# A  Pre-training Quantum Foundation Models

While it remains an open question which quantum labels are best suited for pre-training, we believe there is a strong argument for pre-training on energy. DFT computes quantum properties by solving the optimization problem:

$$\text{energy}^* = \min_{\text{electron\_density}} \text{Energy}[\text{ electron\_density }].$$

All the quantum properties are then derived from the electron density. Pre-training the foundation model to predict energy thus asks the foundation model to perform the same task as DFT. It remains an open problem whether combining energy predicting with other tasks for pre-training would lead to better performance.[10]

**Self-supervision.**  Self-supervised NLP bypass the need for labels $y$ by predicting parts of the input $x$ from $x$. This is useful because it circumvents creating $y$ which require expensive human annotation. In Quantum Chemistry, $x$ is a molecule represented by atom positions and atom types, while $y$ is a quantum chemical property. We note two difference to NLP

1. In Quantum Chemistry $y$ may be computed with DFT and require no human annotation.
2. If the input $x$ arrises from classical chemistry simulations $x$ contains no quantum information. If $x$ arrises from quantum chemistry (e.g. PCQ), the compute required to get $y$ is around 1% of the compute required to compute $x$.

---

[10]Note: Our SchNet experiments trained on gap between molecular orbital energies (HLgap).

