# 1 Datasheet for QM1B

As recommended by the NeurIPS dataset and benchmark track, we documented QM1B and intended uses through the Datasheets for Datasets framework [1]. The goal of dataset datasheets as outlined by [1] is to provide a standardized process for documentating datasets. The authors of [1] present a list of carefully selected questions which dataset authors should answer. We hope our answers to these questions will facilitate better communication between us (the dataset creators) and future users of QM1B.

## 1.1 Motivation

### 1.1.1 For what purpose was the dataset created?

**Purpose.** Prior gaussian-based Density Functional Theory (DFT) datasets contained fewer than 20 million training examples. We suspected the comparatively small dataset sizes held back the performance of neural networks. The purpose of QM1B was thus to present a dataset which increased the number of training examples. We believe this will facilitate novel research directions that train quantum chemical neural networks on more data, e.g., neural scaling laws for quantum chemistry.

**Non-goals.** QM1B should **not** be used to benchmark deep learning architectures. It remains unknown whether the ranking of neural architectures on QM1B will agree with the ranking on experimental datasets or higher resolution DFT datasets. Without evidence for such transferability, benchmarking on QM1B may bias the selection of neural architecture towards our lower resolution DFT. We instead encourage researchers to pretrain on QM1B and finetune/benchmark on higher-resolution DFT datasets like QM9 or PCQ.

### 1.1.2 Who created the dataset and on behalf of which entity?

Alexander Mathiasen and Hatem Helal created the dataset. The dataset was created as part of their employment as Research Scientists at Graphcore. The dataset generation process iteratively proceeded with Mathiasen generating a dataset and Helal training a model. During this process Mathiasen would alter generation parameters (which SMILES strings from GDB, the number of conformers, heavy atoms, basis set) and also optimize the $PySCF_{IPU}$ generation code, while Helal might fix bugs in the SchNet training code and improve the dataloader. This iteration cycle allowed feedback on the DFT dataset generation and DFT library from training SchNet on the generated data. The goal of this process was to bias QM1B towards training neural networks. It is possible that this process further biased QM1B towards SchNet.

### 1.1.3 Who funded the creation of the dataset?

The costs associated with the dataset (compute and salaries) were funded by Graphcore.

## 1.2 Composition

### 1.2.1 What do the instances that comprise the dataset represent (e.g. documents, photos, people, countries)?

The instances (rows) of the dataset represent molecules. Each $N$-atom molecule is represented by its atom positions $\mathbf{R} \in \mathbb{R}^{N \times 3}$, atom types $Z \in \mathbb{N}^N$ and its SMILES string.

### 1.2.2 How many instances are there in total?

The dataset contains 1.07B training examples. Of these there are 1.09M unique molecules. Each unique molecule has up to 1000 different atom positions (conformers) generated using the ETKDG algorithm [2] as implemented in RDKit [3].

### 1.2.3 Does the dataset contain all possible instances or is it a sample of instances from a larger set?

The molecules used to create QM1B are a subset of a particular version of the Generated Data Bank (GDB). There are three published versions of GDB, each named by the maximum number of heavy atoms included: GDB11, GDB13 and GDB17. QM1B contains 1.09M molecules from GDB11 with 9,10,11 atoms from $\{C, N, O, F\}$. GDB11 represents molecules as SMILES strings without explicitly including hydrogens. We added hydrogens using RDKit. The molecules in QM1B contains at most 14 hydrogen atoms.

**Is the sample (QM1B) representative of the larger set (GDB11).** No, the subset of QM1B was chosen to increase the number of training examples by reducing the DFT computation time. This was done by limiting the number of hydrogens to $\leq 14$. This choice was made after visualizing a histogram of the number of hydrogens for GDB11, which, on a log-scaled y axis looks like a bell curve centered around 14 (with at most 25 hydrogens).

**GDB11 as a subset of chemical space.** GDB11 covers a subset of the entire chemical space estimated to contain $10^{60}$ molecules. The subset was filtered for properties like chemical stability [4]. QM1B likely inhereted the biases introduced by the filtering used in GDB11.

### 1.2.4 What data does each instance consist of?

Each row/instance of QM1B contains the input and output of a DFT evaluation:

$$f(atom\_positions, atom\_types) = (HOMO, LUMO, energy).$$

Our 9M SchNet baseline neural network was trained to predict $HLGap = |HOMO - LUMO|$. Each row also contains the SMILES string representation of the molecule from GDB11.

We now explain what each key of the Parquet schema represent:

- 'smile': The SMILES string taken from GDB11. There are up to 1000 rows with the same SMILES string.

- 'atoms': This is a string that represents the atom symbols of the molecule, e.g. "COOH".

- 'z': Integer representation of 'atoms' used by SchNet (the atomic numbers).

- 'energy': energy of the molecule computed by PySCF$_{\text{IPU}}$ (unit eV).

- 'homo': The energy of the Highest Occupied Molecular Orbital (HOMO) (unit eV).

- 'lumo': The energy of the Lowest occupied Molecular Orbital (LUMO) (unit eV).

- 'N': The number of atomic orbitals for the specific DFT computaiton (depends on the basis set STO3G).

- 'std': The standard deviation of the energy of the last five iterations of running PySCF$_{\text{IPU}}$, used as convergence criteria $std < 0.01$ (unit eV).

- 'y': The HOMO-LUMO Gap (unit eV).

- 'pos': The atom positions (unit Bohr).

### 1.2.5 Is there a label or target associated with each instance.

See above. Each row/instance has three potential labels/prediction-targets: (homo, lumo, energy). We performed analysis on the numerical error of each and found energy to have larger numerical errors.

### 1.2.6 Is any information missing from individual instances?

No.

### 1.2.7 Are relationships between individual instances made explicit?

Yes. Each row corresponds to a conformer of a molecule. Each molecule has up to 1000 conformer. This relationship is made explicit by including the SMILES string for each instance (row), that is, up to 1000 rows share the same SMILES string. The validation split we include is made on SMILES strings, ensuring that the same molecule does not appear in training and validation set.

### 1.2.8 Are there recommended data splits?

Training and validation: yes. Test: no.

We include a training/validation split as used in our paper to train SchNet with 9M parameters. We advise against using QM1B as a benchmark for training neural networks due to its low resolution, and thus do not publish a test set.

The main reason for excluding a test set, is to discourage using QM1B to benchmark machine learning models. It remains unclear whether ranking model performance on our "low resolution DFT" will translate to "higher resolution DFT" and experimental measurements. It is thus possible that benchmarking deep learning models on QM1B would lead to a ranking that biases models towards "low resolution DFT" away from experimental measurements. We encourage research into investigating the utility of QM1B for such benchmarking, however, before sufficient evidence is provided, we discourage the use of QM1B as a benchmark to rank machine learning models. Instead, we encourage pretraining on QM1B and fine-tuning/benchmarking on downstream tasks like QM9 or PCQ.

We will communicate the state of such evidence and evolving best-practices through github.com/graphcore-research/qm1b

### 1.2.9 Are there any errors, sources of noise, or redundancies in the dataset?

QM1B contains numerical errors due to the use of float32 instead of float64. For an analysis of the numerical errors of (HOMO, LUMO, Energy, HLGap) please see Figure 2 of the main paper. While the errors of (HOMO, LUMO, HLGap) were below that of neural networks, the numerical errors on energy were similar to that of well-trained neural networks. We thus advise caution when using QM1B to train neural networks for energy prediction. We plan to further optimize the numerical accuracy of $PySCF_{IPU}$ and may in the future recompute labels with improved numerical accuracy.

### 1.2.10 Is the dataset self-contained, or does it link to or otherwise rely on external resources (e.g., websites, tweets, other datasets)?

The QM1B dataset is entirely self-contained.

### 1.2.11 Does the dataset contain data that might be considered confidential (e.g., data that is protected by legal privilege or by doctor– patient confidentiality, data that includes the content of individuals' non-public communications)?

No. The data contains molecules which are not considered confidential.

### 1.2.12 Does the dataset contain data that, if viewed directly, might be offensive, insulting, threatening, or might otherwise cause anxiety?

No.

## 1.3 Collection Process

### 1.3.1 How was the data associated with each instance acquired?

Directly computed using our DFT library $PySCF_{IPU}$ which we open-source. The molecules were obtained from GDB11 for which we got atom positions from RDKit. We validated $PySCF_{IPU}$ against PySCF on 10k molecules from the validation set of QM1B, see Figure 2.

### 1.3.2 What mechanisms or procedures were used to collect the data (e.g., hardware apparatuses or sensors, manual human curation, software programs, software APIs)?

The data was collected using computational chemistry tools run on CPUs and IPUs. In particular, we relied on RDKit and our DFT implementation PySCF$_{\text{IPU}}$. Subsequent filtering were handled manually. We filtered away instances with HLGap 0 and used the standard-deviation of the last 5 Kohn-Sham DFT iterations as a metric to test for convergence. That is, let $energies.shape = (20,)$ be a vector with the energy of all 20 DFT iterations. We then remove molecules for which

$$np.std(energies[-5:]) > 0.01$$

This removed fewer than 1% of the examples. These molecules could be added by falling back to PySCF in float64, we invite pull requests with such fixes.

### 1.3.3 If the dataset is a sample from a larger set, what was the sampling strategy (e.g., deterministic, probabilistic with specific sampling probabilities)?

There was no randomness in the sampling process from GDB11. We explicitly biased the molecules towards fewer number of hydrogen atoms to increase the size of the dataset. Users of the dataset may view exactly which SMILES strings we used by comparing the SMILES strings of GDB11 against QM1B.

### 1.3.4 Who was involved in the data collection process (e.g., students, crowdworkers, contractors) and how were they compensated (e.g., how much were crowdworkers paid)?

Alexander Mathiasen and Hatem Helal was involved in the data collection process. Both were paid through their employment at Graphcore.

### 1.3.5 Over what timeframe was the data collected.

The DFT calculations were started and stopped at different times due to hardware allocation. In particular, the 10 heavy atoms computations where started on May 23 and stopped May 26. The 11 heavy atoms where started on May 25 and, due to an error, continued generating longer than planned (until June 1) generating 3-4x too many training examples for 11 heavy atoms. [1] Finally, the 9 heavy atoms calculation started 26th of May and finished the 29th of May.

The 5 days generation time reported in the main article was calculated as follows: compute the total IPU hours by summing up log files, <40 000, and divide by 320 IPUs.

### 1.3.6 Were any ethical review processes conducted (e.g., by an institutional review board)?

No.

## 1.4 Preprocessing/cleaning/labeling

### 1.4.1 Was any preprocessing/cleaning/labeling of the data done (e.g., discretization or bucketing, tokenization, part-of-speech tagging, SIFT feature extraction, removal of instances, processing of missing values)?

Yes. Hatem Helal and Alexander Mathiasen performed the following manual preprocessing/cleaning.

- Remove molecules with $HLGap = 0$

- Remove unconverged molecules as described above.

- Due to a software error in PySCF$_{\text{IPU}}$ some molecules where run twice. Such duplicates were manually removed.

---

[1]For example, IPU number 27 generated 5.84M training examples and we only used the first 1.5M for QM1B of which 1.47M converged.

### 1.4.2 Was the "raw" data saved in addition to the preprocessed/cleaned/labeled data (e.g., to support unanticipated future uses)?

For convenience QM1B does not include the removed molecules (unconverged and $HLgap = 0$). If anyone is interested, please open a GitHub Issue and we'll release the data for everyone.

### 1.4.3 Is the software that was used to preprocess/clean/label the data available?

Our postprocessing code is based on the open-source libraries: pandas, pyarrow and pqdm. We intend to improve/rewrite our postprocessing pipeline and thus decided against making it available in the first release.

## 1.5 Uses

### 1.5.1 Has the dataset been used for any tasks already?

Yes. We used QM1B to train the neural network SchNet 9M to predict HOMO-LUMO Gap.

### 1.5.2 Is there a repository that links to any or all papers or systems that use the dataset?

No .

### 1.5.3 What (other) tasks could the dataset be used for?

The dataset could be used for pretraining large models subsequently fine-tuned on down-stream tasks. The possible prediction tasks are any combination of HOMO, LUMO and total energy. It might be possible that the scale of QM1B allows infering forces from energies[2], however, the larger numerical errors seem to make this unlikely.

### 1.5.4 Is there anything about the composition of the dataset or the way it was collected and preprocessed/cleaned/labeled that might impact future uses?

The lower DFT resolution. We currently advise against using QM1B to benchmark machine learning models. Instead, we encourage pretraining on QM1B and subsequently fine-tuning on downstream tasks like PCQ and QM9.

### 1.5.5 Are there tasks for which the dataset should not be used?

Currently we advise against using QM1B to benchmark machine learning models. It remains unclear whether ranking machine learning models on QM1B would translate to experimental measurements. We instead encourage pretraining on QM1B and benchmarking models on downstream tasks on QM9 or PCQ.

## 1.6 Distribution

### 1.6.1 Will the dataset be distributed to third parties outside of the entity (e.g., company, institution, organization) on behalf of which the dataset was created?

Yes. The dataset was created to be shared openly. We hope to iteratively refine the current dataset, and invite the entire research community to publicly share any critiques and participate in the iterative improvement of future versions of QM1B. The discussion will be facilitated through GitHub issues on github.com/graphcore-research/qm1b.

### 1.6.2 How will the dataset be distributed (e.g., tarball on website, API, GitHub)?

The dataset is stored in multiple Apache Parquet files. These are initially available to download from Amazon S3 and longer term storage is being arranged on the Figshare platform. We will also provide a programming API to automate downloading the dataset to facilitate training neural networks.

---

[2]Recall that the forces acting on nuclei is the negative of the derivative of energy with respect to atom positions.

### 1.6.3 Will the dataset be distributed?

Yes. As mentioned above, the dataset is initially made available through Amazon S3 and long term storage is being arranged on the Figshare platform. We plan to release the dataset to the public through Figshare before the NeurIPS conference.

### 1.6.4 Will the dataset be distributed under a copyright or other intellectual property (IP) license, and/or under applicable terms of use (ToU)?

QM1B will be released under the same license as GDB11 Creative Commons Attribution 4.0 International.

### 1.6.5 Have any third parties imposed IP-based or other restrictions on the data associated with the instances?

No.

### 1.6.6 Do any export controls or other regulatory restrictions apply to the dataset or to individual instances?

No.

## 1.7 Maintenance

### 1.7.1 Who will be support/hosting/maintaining the dataset?

**Support.** We will provide support for QM1B by answering GitHuB issues.

**Hosting.** QM1B will be hosted on Figshare and the $875 long-term hosting fee will be paid by Graphcore.

**Maintenance.** The maintenance of dataset updates and corrections will be facilitated through the GitHub repository github.com/graphcore-research/qm1b. We invite users to supply any feedback, questions, and bug reports as GitHub Issues. We will document corrections and updates through the CHANGELOG and provide versioned releases for any major updates.

### 1.7.2 How can the owner/curator/manager of the dataset be contacted (e.g., email address)?

Raise a GitHub issues through github.com/graphcore-research/qm1b.

### 1.7.3 Is there an erratum? If so, please provide a link or other access point.

The CHANGELOG on github.com/graphcore-research/qm1b.

### 1.7.4 Will the dataset be updated (e.g., to correct labeling errors, add new instances, delete instances)? If so, please describe how often, by whom, and how updates will be communicated to dataset consumers (e.g., mailing list, GitHub)?

QM1B will be updated in case of any human error in the postprocessing steps, otherwise, we have no plans to update QM1B. Our main focus for updates will be on extending the capabilities of $PySCF_{IPU}$ to allow the community to create larger and more accurate datasets.

How will updates be communicated? Discussion regarding updates will be facilitated through Github issues while documentation of updates will be communicated through the CHANGELOG.

### 1.7.5 If the dataset relates to people, are there applicable limits on the retention of the data associated with the instances (e.g., were the individuals in question told that their data would be retained for a fixed period of time and then deleted)? If so, please describe these limits and explain how they will be enforced.

QM1B contains molecules and does not relate to people.

### 1.7.6 Will older versions of the dataset continue to be hosted/maintained? If so, please describe how. If not, please describe how its obsolescence will be communicated to dataset consumers.

In case of updates we will only release "diffs" ensuring the initial version of the dataset remains available.

### 1.7.7 If others want to extend/augment/build on/contribute to the dataset, is there a mechanism for them to do so?

Yes. Create a GitHub fork of github.com/graphcore-research/qm1b. Upload the additional dataset on Figshare. Change the github.com/graphcore-research/qm1b/README.md with a 3-10 line python example show-casing how to load the additional data. (Optional) Create a pull-request to the README of github.com/graphcore-research/qm1b with a link to your fork.