# OpenReview forum: "Generating QM1B with PySCF$_{\text{IPU}}$"
_NeurIPS.cc/2023/Track/Datasets_and_Benchmarks — NeurIPS 2023 Datasets and Benchmarks Poster_

### Official Review · Reviewer_mTsd · 2023-07-16
**A good work creating a useful large datasets for training foundation models in chemistry**

**Rating:** 7
**Confidence:** 4
**Correctness:** Yes, the dataset is constructed in a …
**Clarity:** Yes.

**Strengths:**

This work presents a very interesting and useful large-scale dataset for molecular machine learning community. The size of the presented dataset is the largest to date in the community. It will be broadly useful to researchers in the community to evaluate the performance or dig the potential of large-scale molecular machine learning models.

**Additional Feedback:**

No additional feedback.

**Documentation:**

Yes.

**Ethics:**

No ethical concerns.

**Limitations:**

Yes.

**Opportunities For Improvement:**

Major:
As the presented QM1B dataset is mainly used for developing foundation models, authors are recommended to do the following improvements:
(1) Add some explanations or discussions about why predicting molecular potential energies is a good task for training foundation models, instead of possibly coming up with some self-supervised learning tasks as they are commonly used in NLP community.
(2) Conduct experiments with some larger transformer-based models (e.g., Graphormer) on QM1B datasets, and compare the performance difference between models with different sizes.

Minor:
As float32 is used to accelerate the data generation process but causes higher errors, authors are recommended to add explanations or discussions about the time cost of using float64.

**Relation To Prior Work:**

Yes.

**Summary And Contributions:**

This work presents QM1B, a large dataset with more than 1 billion data samples for molecular potential energy prediction. QM1B is the first molecule dataset produced by running PySCF on Intelligence Processing Units. SchNet is observed to achieve performance gain after training on QM1B dataset.

---

> ### Author Response · Authors · 2023-08-22
> **Review mTsd (7)**
>
> Thank you for spending your valuable time reviewing our work. We commend the reviewer for appreciating the size of our dataset, and its broad usefulness to researchers and its potential for large-scale molecular machine learning models.
>
>
> > Add some explanations or discussions about why predicting molecular potential energies is a good task for training foundation models, instead of possibly coming up with some self-supervised learning tasks as they are commonly used in NLP community.
>
> Thank you for raising this important question. The following argues that pre-training foundation models differs between NLP and molecules. In self-supervised NLP, masking allows us to circumvent training on (x,y) by predicting x from parts of x. This is useful because it circumvents creating y which requires expensive human annotations. In molecular machine learning x may be atom positions or SMILES strings constructed using
> 1. classical chemistry simulations
> 2. quantum chemistry simulations
>
> If x is constructed using classical chemistry (e.g. GDB11) there is no quantum information in x. In other words: while predicting atom 5 from all other atoms may be useful for pre-training foundation models for classical chemistry, we are uncertain how this could significantly improve performance on downstream quantum tasks (e.g. the prediction tasks in QM9). Alternatively, one may generate $x$ using quantum chemistry (e.g. PCQ). In this case, predicting atom 5 from all the other atom positions does contain information about quantum chemistry, however, the labels $y$ (energy, HLGap, ...) come for free as they are a by-product of computing $x$. **In summary:** the advantage of self-supervision in NLP is circumventing the expensive human labeling $y$, meanwhile, QM1B without $y$ is "just" GDB11 and contains no quantum information.
>
> The reviewer raised another interesting point. Of the potential labels $y$ one could compute, why use molecular potential energy? While it remains an open question which quantum labels $y$ are best suited for pre-training, we believe there is a strong argument for pre-training on energy. DFT computes quantum properties $y$ by solving the optimization problem:
> $$
> \quad\quad\quad\quad \text{energy}^*=\min_{\text{electron density}} \text{energy}(\text{electron density})
> $$
> All the quantum properties $y$ are then derived from the electron density. Pre-training the foundation model to predict energy thus asks the foundation model to perform the same task as DFT. It remains an open problem whether combining energy predicting with other tasks for pre-training would lead to better performance.
>
> We will update any final version with a subsection titled "Pre-training Quantum Foundation Models" outlining the above argument. Please let us know if you think there's anything we could further improve.
>
> > Conduct experiments with some larger transformer-based models (e.g., Graphormer) on QM1B datasets, and compare the performance difference between models with different sizes.
>
> Thank you for raising this point, we ourselves believe this is an intriguing avenue for future work. Taking all our reviews into consideration, we decided to prioritize our pre-training/fine-tuning experiment (we managed to improve SchNet MAE HLGap on QM9 from **54.13meV** to **30.2meV** simply by pre-training on QM1B). This result may indicate that prior work successfully addressed neural architectures, but may have under appreciated the importance of simply scaling dataset size. Indeed, interpreting our result through Table 5 in [44], we see that SchNet, a 7 year old model, can be improved from the top 17 (second worst) to top 5 simply by pre-training.

---

> > ### Comment · Reviewer_mTsd · 2023-08-22
> > **Response**
> >
> > I appreciate authors' work to address my major concerns. I encourage authors to address my remaining minor concern in the review, i.e., adding explanations or discussions about what is the time cost of using float64.

---

> > > ### Author Response · Authors · 2023-08-22
> > >
> > > Apologies, please find our reply below.
> > >
> > > > Minor: As float32 is used to accelerate the data generation process but causes higher errors, authors are recommended to add explanations or discussions about the **time cost** of using float64.
> > >
> > > Our understanding, is that machine learning hardware accelerators like Cerebras, TPUs and IPUs do not support float64 compute instructions in hardware (float64 is rarely used in deep learning). For our use case, we are left with the option to simulate float64 in software. This was also the case for the authors of [this article](https://arxiv.org/pdf/2202.01255.pdf) who report a ~11x degradation when simulating float64 in software for their DFT on TPUs. Following your question, we ran a rough float64 experiment for the electron repulsion integrals (we simply changed casting from float to double in `int2e_sph.cpp`). This causes LLVM compilation to simulate double precision operations, which, we found increased running time by ~6x. We'll update any final version with a paragraph detailing a more thorough experiment. Please do let us know if any of the above leads to further questions.
> > >
> > > Related to the "higher [numerical] errors", you might find our reply to another reviewer interesting. We copied the answer below for your convenience.
> > >
> > > > We believe float32 DFT is an important area for further research. Inspired by our reviews we performed a pretrain/fine-tuning experiment, which we believe demonstrates that float32 is sufficient to generate data with DFT specifically for pre-training when subsequently fine-tuning on a small accurate dataset (like QM9/float64). Our experiment improved SchNet HLGap MAE from **54.13meV** (trained on QM9/float64) to **30.2meV** (pre-trained on QM1B/float32 then fine-tuned QM9/float64).
> > >
> > > > **Note:** It took a community-wide effort to utilize lower precision in deep learning, it may require a similar amount of work to fully utilize lower precision for DFT. While PySCF-IPU demonstrates the utility of float32 for the particular use-case of dataset generation, its current numerical precision may be inadequate for other tasks. We hope open-sourcing PySCF-IPU will accelerate research into lower precision DFT, in particular, we ourselves found the following features helpful:
> > >
> > > > 1. Experiments of the form: "Compute DFT with all operations in float64 but keep one operation in float32" [see here](https://github.com/graphcore-research/pyscf-ipu/blob/main/density_functional_theory.py#L1365).
> > > > 2. Visualize numerics of intermediate tensors through each DFT iteration, [see animation here](https://raw.githubusercontent.com/graphcore-research/pyscf-ipu/main/images/visualize_DFT_numerics.gif).
> > >
> > > > These above features allowed us to achieve **MAE(float32, float64)=0.2meV** (Figure 2), lower than the error between DFT datasets due to different DFT options **MAE(PCQ,QM9)=81.14meV**.

---

> > > > ### Comment · Reviewer_mTsd · 2023-08-22
> > > > **Response**
> > > >
> > > > Thanks for your response. All my concerns have been addressed now. I will keep my rating.

---

### Official Review · Reviewer_hHv3 · 2023-07-22
**A review for 759**

**Rating:** 6
**Confidence:** 3
**Correctness:** The paper has no problems with correc…

**Strengths:**

1. The data generation framework proposed by the authors is based on python and implements part of the computation in c. It also possesses hardware-friendly adaptations for IPUs, which can significantly improve the speed of data construction. It also allows users to configure the implementation details, which greatly facilitates the construction of datasets in related fields.
2. By constructing a simple neural network trained on subsets of the proposed dataset of different sizes, the results show that the model performance continues to improve as the dataset continues to grow, which demonstrates the research significance of this work.


**Additional Feedback:**

See strength and weakness.

**Clarity:**

The overall structure of the paper is clear and the writing is logical and relatively easy to understand and read.

**Documentation:**

In section 3, the authors detail the framework used to generate the dataset and provide open source code for the framework, making it possible to autonomously build datasets that meet individual needs. In section 4, the authors further provide a detailed description of the constructed dataset, including its characteristics and validity.

**Ethics:**

No ethic problem.

**Limitations:**

As mentioned before, the QM1B dataset could not be applied to practical pharmaceutical research because of its lack of precision.

**Opportunities For Improvement:**

1. The paper gives the impression that the core work is mainly in the proposed data generation framework rather than the constructed dataset. The paper lacks some introduction of further characterization of the proposed QM1B dataset and implementation of application scenarios.
2.  The authors constructed the dataset in their paper in such a way as to increase the amount of data obtained by decreasing the precision. This makes this dataset potentially difficult to apply in real pharmaceutical research, and possesses greater limitations in its application. In fact, since there is a compensatory relationship between precision and scale, the authors can further explore how to trade off the two to make the whole dataset satisfactory in terms of scale, but also have a precision that can be used for rigorous research.


**Relation To Prior Work:**

The authors provide a detailed description and comparison of related work in their paper. And some of the results of people applying DFT methods before that are detailed.

**Summary And Contributions:**

This paper present a new hardware accelerated DFT data generator for deep learning in the field of molecular machine learning, which is an improvement for the original CPU-based DFT method. And based on this method, a dataset with low resolution but possessing a size of 109 was constructed. In order to verify the validity of the dataset, a simple neural network is proposed, whose performance gets better as the amount of data increases.

---

> ### Author Response · Authors · 2023-08-21
> **Reviewer hHv3 (6)**
>
> Thank you for spending your valuable time reviewing our work. We commend the reviewer for appreciating our data generator can accelerate dataset creation, not just for QM1B, but potentially other related fields.
>
> > The paper lacks some introduction of further characterization of the proposed QM1B dataset and implementation of application scenarios.
>
> Following our reviews we focused on characterizing QM1B's application scenarios. In particular, we pre-trained SchNet on QM1B and subsequently fine-tuned on the application scenario QM9 (we chose QM9 due to its usage throughout the literature). Pre-training/fine-tuning improved the HLGap MAE of SchNet from **54.13meV** to **30.2meV**. We will update any final version with a paragraph in Section 4 titled *“An Application Scenario: Pre-training on QM1B and fine-tuning on QM9”* detailing this experiment.
>
> To further improve QM1B's characterization, we added the following to GitHub: a [dataset README](https://github.com/graphcore-research/pyscf-ipu/tree/main/qm1b) and a "[Datasheet for Datasets](https://github.com/graphcore-research/pyscf-ipu/blob/0c9cbf4f2312c7f7d95ca0292fb163b406434238/qm1b/datasheet.pdf)" (as recommended by the [guidelines](https://neurips.cc/Conferences/2023/CallForDatasetsBenchmarks)). Please let us know if you can think of any further actions that would improve our characterization.
>
> > The authors constructed the dataset in their paper in such a way as to increase the amount of data obtained by decreasing the precision. This makes this dataset potentially difficult to apply in real pharmaceutical research, and possesses greater limitations in its application. In fact, since there is a compensatory relationship between precision and scale, the authors can further explore how to trade off the two to make the whole dataset satisfactory in terms of scale, but also have a precision that can be used for rigorous research.
>
> Thank you for raising this critical point.  Our main inspiration stems from the recent progress in natural language processing and computer vision: pre-train on large quantities of less accurate data, then fine-tune on small amounts of high-accuracy. We envision QM1B to facilitate such pre-training for molecules. We believe our pre-train/finetune experiment (see above) demonstrates that pre-training on QM1B improves accuracy on applications like QM9. This opens the possibility that, while QM1B will not replace high accuracy pharmaceutical datasets, it may improve the performance of models on pharmaceutical datasets by allowing large scale pre-training.
>
> Finally, we hope to facilitate research that investigates the important trade-off between accuracy and scale
> by open-sourcing our data generator [PySCF IPU](https://github.com/graphcore-research/pyscf-ipu).
>
> > The paper gives the impression that the core work is mainly in the proposed data generation framework rather than the constructed dataset.
>
> This impression was intended by the authors. While we believe QM1B will accelerate the development of molecular machine learning through pre-training, we hope our data generator PySCF-IPU will have an even bigger impact by allowing the deep learning community to continuously improve on molecular dataset generation.
>
> We hope our answers address some of the reasons the reviewer may have chosen to rate the paper "Marginally above acceptance threshold", and would like to ask if this rating might be reconsidered in the light of this new information.

---

> > ### Comment · Reviewer_hHv3 · 2023-08-29
> >
> > Thanks for the authors effort on their rebuttal. They partially address my concerns. However, I still have concerns on the decreased precision of the dataset. Therefore, I decide to keep my rating.

---

### Official Review · Reviewer_Aoip · 2023-07-24
**Large Scale Quantum Chemistry dataset with Approx. DFT and Better Hardware**

**Rating:** 8
**Confidence:** 3
**Correctness:** There are no correctness issues with …
**Clarity:** Paper is clearly written

**Strengths:**

- The dataset provided is significantly larger than previous datasets for the tasks.
- Baseline model (Schnet) gives significantly improved predictions.
- The training and validation errors seem to converge at the dataset size, indicating that further improvement might not be possible with larger datasets.

**Additional Feedback:**

comments mentioned above.

**Documentation:**

The code was provided as a zip file. It was better if a proper website, with detailed documentation was provided.

**Ethics:**

There are not ethics issues with the problem/dataset.

**Limitations:**

- The use of float32 instead of float64 in the IPU code seems to affect the accuracies. This needs to be understood better.

**Opportunities For Improvement:**

The paper is well written

**Relation To Prior Work:**

Prior work is sufficiently mentioned and differentiated.

**Summary And Contributions:**

The paper presents the QM1B dataset for Quantum chemistry with 1 Billon structures of 9-11 heavy atoms. The dataset has more examples by a large factor compared to previous datasets. The dataset is obtained using faster DFT approximations that give less accurate data and better computing setup (IPUs). A baseline model (SchNet) shows improved Mean Absolute error (>200 Mev) on property predictions on the larger dataset.

---

> ### Author Response · Authors · 2023-08-21
> **Reviewer Aoip (8)**
>
> Thank you for spending your valuable time reviewing our work. We commend the reviewer for appreciating that our dataset QM1B is significantly larger than prior datasets and that we used a better computing setup (IPUs).
>
> > The code was provided as a zip file. It was better if a proper website, with detailed documentation was provided.
>
> Inspired by your suggestion we improved our documentation as follows:
>
> 1. Open-sourced [PySCF-IPU on Github](https://github.com/graphcore-research/pyscf-ipu).
> 2. Documented DFT within 300 lines of code in [nanoDFT](https://github.com/graphcore-research/pyscf-ipu/blob/main/nanoDFT/nanoDFT.py). The documentation strives to use deep learning terminology instead of physics terminology, our aim is to provide a self-contained introduction to DFT for deep learning experts.
> 3. Created [two Jupyter notebooks](https://github.com/graphcore-research/pyscf-ipu/tree/main/notebooks) which run our code on IPUs with one click in the [Paperspace cloud](https://ipu.dev/ipobmC).
>
> Please do let us know if you find any aspects of the documentation which we can further improve.
>
> > The use of float32 instead of float64 in the IPU code seems to affect the accuracies. This needs to be understood better.
>
> We agree. We believe float32 DFT is an important area for further research. Inspired by our reviews we performed a pretrain/fine-tuning experiment, which we believe demonstrates that float32 is sufficient to generate data with DFT specifically for pre-training when subsequently fine-tuning on a small accurate dataset (like QM9/float64). Our experiment improved SchNet HLGap MAE from **54.13meV** (trained on QM9/float64) to **30.2meV** (pre-trained on QM1B/float32 then fine-tuned QM9/float64).
>
> **Note:** It took a community-wide effort to utilize lower precision in deep learning, it may require a similar amount of work to fully utilize lower precision for DFT. While PySCF-IPU demonstrates the utility of float32 for the particular use-case of dataset generation, its current numerical precision may be inadequate for other tasks. We hope open-sourcing PySCF-IPU will accelerate research into lower precision DFT, in particular, we ourselves found the following features helpful:
>
> 1. Experiments of the form: "Compute DFT with all operations in float64 but keep one operation in float32" [see here](https://github.com/graphcore-research/pyscf-ipu/blob/main/density_functional_theory.py#L1365).
> 2. Visualize numerics of intermediate tensors through each DFT iteration, [see animation here](https://raw.githubusercontent.com/graphcore-research/pyscf-ipu/main/images/visualize_DFT_numerics.gif).
>
> These above features allowed us to achieve **MAE(float32, float64)=0.2meV** (Figure 2), lower than the error between DFT datasets due to different DFT options **MAE(PCQ,QM9)=81.14meV**.

---

### Official Review · Reviewer_NGMZ · 2023-07-28

**Rating:** 6
**Confidence:** 3

**Strengths:**

- Unclear.

**Additional Feedback:**

I do not understand what the value is of this dataset given the limited accuracy. Unless I can understand that this accuracy is sufficiently good for downstream tasks I see no need to accept this paper.

A benchmark should be providing a clear benefit to the community. As the paper notes, it is unclear whether the dataset is sufficiently accurate.

**Clarity:**

- Overall the paper is easy to read.

**Correctness:**

- Unknown due to experiments not provided.

**Documentation:**

- Unclear due to proprietary data.

**Ethics:**

Unclear.

**Limitations:**

- The dataset is large but as noted in the paper the practical implication of the reduced accuracy is unknown. Therefore the paper recommends this dataset to be only used for research. The paper views this as future research but I believe that this paper should demonstrate that the dataset has a practical use.

- If I understand the paper correctly, the errors made in the data generation are much larger than the errors made by a typical ML model (Schnet in this case).

- The data is generated on proprietary hardware.

**Opportunities For Improvement:**

- Evaluate whether the models trained on this dataset can replace models trained on smaller but more accurate datasets in practice.

**Relation To Prior Work:**

- Limited. It is not clear whether this dataset can

**Summary And Contributions:**

The paper proposes to generate a very large dataset for quantum chemistry predictions.
The paper notes that this dataset uses lower precision than what is commonly done in the field and that the practical implications of this design choice are unknown.The paper leaves this as future work.

---

> ### Author Response · Authors · 2023-08-15
> **Reviewer NGMZ (4)**
>
> Thank you for spending your valuable time reviewing our work. Please find our answers below.
> > Unless I can understand that this accuracy is sufficiently good for downstream tasks I see no need to accept this paper.
>
>
> Inspired by your comments, we pretrained SchNet on QM1B and fine-tuned it on the downstream task QM9.  **This improved HLGap MAE from 54.13+-0.85meV** (normal training) **to 30.2+-0.33meV** (pretrain+finetune). This leads us to believe the accuracy of QM1B is sufficiently good for downstream tasks like QM9.
>
> **Note:** We report mean+-std over three repetitions. Our 54.13+-0.85meV baseline performs on par with this [paper](https://arxiv.org/pdf/1712.06113.pdf) (see Table 1). To avoid data leakage we finetune/evaluate only on the subset of molecules in QM9 with SMILES strings that do not appear in QM1B.
>
>
> > Opportunities For Improvement: Evaluate whether the models trained on this dataset [QM1B] can replace models trained on smaller but more accurate datasets in practice.
>
>
> Our precise claim is:
>
>     “models [pre]trained on this dataset [and then fine-tuned on smaller but more accurate datasets] are more accurate than models trained on smaller but more accurate datasets”.
>
>
> We will clarify this in any final version.  Inspired by your comments we evaluated the claim (see above).  The pretrained/finetuned model got **30.23meV**, so it can replace the model trained on the smaller but more accurate dataset (which got **54.13meV**).
>
> **Note:** The technique of pre-training and fine-tuning is used extensively in natural language processing and computer vision. Our contribution is a dataset that allows such pre-training for molecules at a previously unprecedented scale.
>
>
> >... , the errors made in the data generation are much larger than the errors made by a typical ML model (Schnet in this case).
>
>
> This statement is true for other DFT datasets. Example: The DFT labels from the datasets QM9 and PCQ disagree.
>
>
>     MAE(QM9, PCQ) = 81.14meV # for homo-lumo gap
>
>
> This error is larger than the one of SchNet on QM9.
>
>
>     MAE(QM9, SchNet) = 54.13meV # for homo-lumo gap
>
>
> Irrespective of data generation errors, prior work found pre-training on PCQ and fine-tuning on QM9 improves performance [34,42,43]. This is in line with our finding that pretraining on QM1B and fine-tuning on QM9 improves performance.
>
>
>
>
> > “Limitation: Data generated on proprietary hardware [IPUs]”:
>
>
> IPUs are available in the same way as CPUs/GPUs/TPUs: they can be purchased directly, or through cloud providers such as [Paperspace](https://www.graphcore.ai/paperspace) or Gcore. Furthermore, our data generator, [PySCF IPU](https://github.com/graphcore-research/pyscf-ipu), is written in JAX and therefore supports CPUs/TPUs/GPUs/IPUs.
>
>
> > “Correctness: Unknown due to experiments not provided”:
>
>
> Our experimental code is available to reviewers (see openreview and supplementary material). We also open-sourced our experimental code on [github](https://github.com/graphcore-research/pyscf-ipu/tree/main/schnet_9m).
>
>
> > “Documentation: Unclear due to proprietary data”:
>
>
> Our data is available to reviewers (see openreview and supplementary material). Our dataset will be made publicly available with long term storage provided by Figshare under CC 4.0 Licence. Our dataset was documented using “[Datasheet for Datasets](https://arxiv.org/pdf/1803.09010.pdf)”, and now a [public Github readme](https://github.com/graphcore-research/pyscf-ipu/tree/main/qm1b). Our choice of licence (creative commons) and documentation (Datasheet for Datasets) both follow the [NeurIPS guidelines](https://neurips.cc/Conferences/2023/CallForDatasetsBenchmarks).
>
>
> We believe this addresses many of the reasons the reviewer may have chosen to rate the paper "Ok but not good enough", and would like to ask if this rating might be reconsidered in the light of this new information.

---

### Author Response · Authors · 2023-08-24
**Discussion Reminder**

Thank you to everyone who has engaged during the discussion period so far. We would be very keen to engage in the discussion with the remaining reviewers before the discussion period ends on Tuesday (29/08/2023). Best regards, the authors

---

### Decision · Program_Chairs · 2023-09-22

**Decision:**

Accept (Poster)

**Comment:**

Paper uses technological advancements (Hardware accelerator, programming frameworks like JAX) and spend their computational resources to deliver a dataset that is orders of magnitude larger than other Quantum Chemistry datasets. With billions of training examples, training simple models (E.g., SchNet, as they demonstrate) gives no difference on performance on train VS test partitions, and SchNet is known to pass the "chemical accuracy".

I hope that future versions of the dataset could contain larger molecules and higher data resolutions (float64 instead / in addition to float32).